# Deductive Beam Search: Decoding Deducible Rationale for Chain-of-Thought Reasoning

**Tinghui Zhu**[♠*†] **Kai Zhang**[♡*] **Jian Xie**[♠] **Yu Su**[♡‡]

[♠]Fudan University  [♡]The Ohio State University

thzhu22@m.fudan.edu.cn, {zhang.13253, su.809}@osu.edu

## Abstract

Recent advancements have significantly augmented the reasoning capabilities of Large Language Models (LLMs) through various methodologies, especially chain-of-thought (CoT) reasoning. However, previous methods often struggle to address reasoning errors in intermediate steps, which can lead to accumulative errors. In this paper, we propose **Deductive Beam Search** (DBS), which seamlessly integrates CoT and deductive reasoning with step-wise beam search for LLMs. Our approach deploys a verifier, verifying the deducibility of a reasoning step and its premises, thus alleviating the error accumulation. Furthermore, we introduce a scalable and labor-free data construction method to amplify our model's verification capabilities. Extensive experiments demonstrate that our approach significantly enhances the base performance of LLMs of various scales (7B, 13B, 70B, and ChatGPT) across 8 reasoning datasets from 3 diverse reasoning genres, including arithmetic, commonsense, and symbolic. Moreover, our analysis proves DBS's capability of detecting diverse and subtle reasoning errors and robustness on different model scales. Data and codes are released at https://github.com/OSU-NLP-Group/Deductive-Beam-Search.

## 1 Introduction

Machine reasoning has witnessed tremendous progress thanks to the emergence of Large Language Models (LLMs) (OpenAI, 2023; Google, 2023; Anil et al., 2023; Touvron et al., 2023; McIntosh et al., 2023). The power of LLMs activates the ability to conduct step-by-step chain-of-thought (CoT) reasoning (Wei et al., 2022b;a), significantly boosting the performance of reasoning tasks (Wang et al., 2022; Paul et al., 2023; Lyu et al., 2023).

Although CoT reasoning has demonstrated the superiority of step-by-step reasoning, its dependency on intermediate steps inevitably introduces accumulative errors

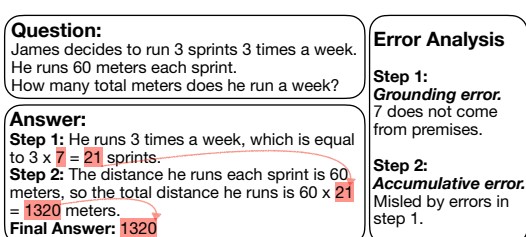

Figure 1: Example of error in an intermediate step leading to accumulative error from *Llama2-7b*. The dependency on intermediate steps introduces accumulative errors in the reasoning process.

(Du et al., 2023; Yu et al., 2023) in the process, as shown in Figure 1. Previous research that alleviates these errors lies in two main paradigms: *1) Answer aggregation across multiple rationales*. They utilize majority voting (Wang et al., 2022) or deploy a verifier to score on each rationale (Li et al., 2023). However, these methods do not directly address errors in the reasoning process, undermining the reliability of their outcomes. *2) Intermediate step*

---

[*]The first two authors contributed equally.

[†]Work done during Tinghui Zhu's internship at OSU NLP Group.

[‡]Corresponding author.

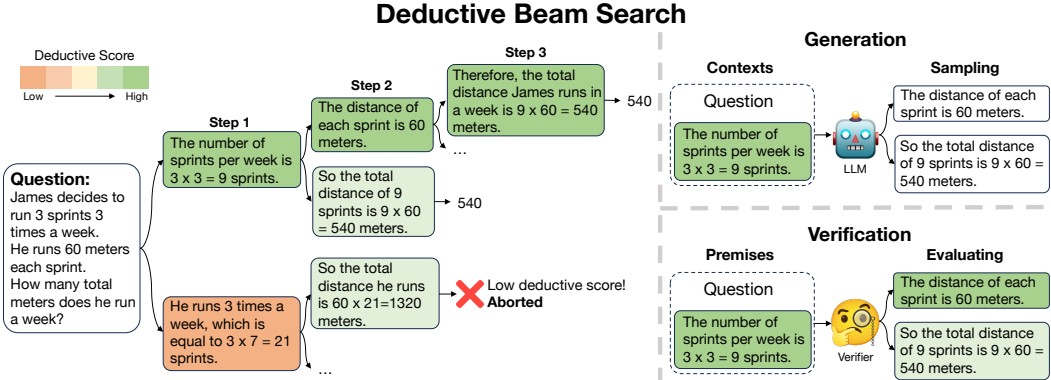

Figure 2: Overview of Deductive Beam Search. We illustrate the process under the configuration of beam size 2 and sampling times 2.

*correction.* This line of works decomposes the reasoning path into reasoning steps and applies self-correction on each step (Weng et al., 2022; Ling et al., 2024; Paul et al., 2023; Xie et al., 2023). Yet, recent research finds that, without external feedback, LLMs tend to modify reasoning steps regardless of their correctness (Huang et al., 2023; Hong et al., 2023).

Previous works fail to address reasoning errors in intermediate steps, compromising the ability to conduct systematic reasoning. To mitigate this issue, we embrace the principle of deductive reasoning (Clark, 1969; Johnson-Laird, 1999; 2010). In deductive reasoning, every step logically follows its premises, where a deducible reasoning step is termed a **logical consequence** (Dinkmeyer, 1976; Hanson, 1997). A key attribute of logical consequence is that if the premises hold, the deducible reasoning step is true, suggesting a correct outcome. Inspired by this attribute, we propose to navigate CoT towards a more deducible path.

Nonetheless, challenges arise when introducing the principle of deductive reasoning into CoT reasoning without changing the standard prompt paradigm and the parameters of LLMs. **1) Navigation on CoT reasoning.** Since LLMs cannot always conduct correct deductive reasoning, they have to explore the potential reasoning space and choose those reasoning steps that are more likely to be deducible, which brings the trade-off between exploration and exploitation (Donoso et al., 2014; Dasgupta et al., 2019). **2) Verification of deducibility.** On one hand, previous research shows it is hard for LLMs to detect reasoning errors (Huang et al., 2023; Hong et al., 2023). On the other hand, symbolic reasoning engines (Cavada et al., 2014; Li et al., 2018) can reliably verify the correctness. However, transferring natural language to symbolic language without losing generality remains an unsolved problem in machine reasoning.

Confronted with these challenges, we propose **Deductive Beam Search** (DBS), adaptable to all models and settings. The overview of DBS is shown in Figure 2. For the trade-off challenge, we decompose the reasoning process into reasoning steps and incorporate step-wise beam search. In terms of the verification challenge, we propose a deductive verifier, which takes a reasoning step and its premises as inputs and outputs a deductive score, evaluating the logical coherence between them. Specifically, LLM samples a list of potential reasoning steps to explore. Then, our deductive verifier exploits by selecting steps that are more deducible. To train an effective verifier, we propose a scalable way of synthesizing fine-grained and diverse deductive reasoning errors without human annotation. Initially, the verifier is trained to verify heuristically synthesized wrong steps with typical reasoning error patterns. Subsequently, we ask LLMs to generate reasoning steps where false ones detected by our verifier serve as hard negatives. These hard negatives are adopted to train a deductive verifier with model feedback.

As we aim to enable LLMs to decode more deducible reasoning paths, DBS can be integrated with answer-aggregation-based methods. We evaluate our methods across 5 arithmetic reasoning tasks, 2 commonsense reasoning tasks, and 1 symbolic reasoning task in single chain setting and multiple chain setting. The improvements can be expected not only on models of all scales and diverse model families but also under different settings. Concretely,

taking arithmetic reasoning tasks as an example, the average improvement is 5.3% / 3.2% on Llama2-7b / ChatGPT under single chain setting and 3.9% / 2.5% under multiple chain setting. Moreover, we comprehensively analyze our verifier, demonstrating its capability of detecting diverse and subtle reasoning errors and robustness on different model scales.

## 2   Deductive Beam Search

We begin by formulating multi-step CoT reasoning with step-wise beam search before describing DBS. For notation convenience, we denote $[n]$ to be a set of natural numbers from 1 to $n$, and $\mathbf{v}_{[n]} = [v_1, v_2, ..., v_n]$ represents the first $n$ elements of $\mathbf{v}$, where $\mathbf{v}_{[0]} = [\,]$ representing an empty sequence. Specifically, we denote tokens as $y$.

### 2.1   Multi-Step Chain-of-Thought Reasoning

Standard chain-of-thought reasoning (Wei et al., 2022b) generates the whole reasoning path for the final outcome. Formally, given the question $\mathbf{q}$, CoT formulates the answer distribution $\Pr_{LM}(a|\mathbf{q})$ as a product of the rationales generation distribution $\Pr_{LM}(\mathbf{r}_{[t]}|\mathbf{q})$ and a final answer distribution $\Pr_{LM}(a|\mathbf{r}_{[t]})$, which is:

$$\Pr_{LM}(a|\mathbf{q}) = \Pr_{LM}(a|\mathbf{r}_{[t]}) \times \Pr_{LM}(\mathbf{r}_{[t]}|\mathbf{q}), \tag{1}$$

where $\mathbf{r}_{[t]} = [\mathbf{r}_1, \mathbf{r}_2, ..., \mathbf{r}_t]$ is a complete reasoning path, and $t$ is the number of steps required to complete the reasoning process. Each $\mathbf{r} = [y_1, y_2, ...y_l]$ is an intermediate reasoning step, where $l$ is its token length.

Problems in this setting lie in the complexity of navigating the generation of $\mathbf{r}_{[t]}$, which are sampled as a whole directly from language models, a process wherein errors can accumulate (Zhang et al., 2023a). To avoid error accumulation and navigate the reasoning process, we decompose the process of generating $\mathbf{r}_{[t]}$ as:

$$\Pr_{LM}(\mathbf{r}_{[t]}|\mathbf{q}) = \Pr_{LM}(\mathbf{r}_1|\mathbf{q}) \times \prod_{i=1}^{t-1} \Pr_{LM}(\mathbf{r}_{i+1}|\mathbf{q}, \mathbf{r}_{[i]}) = \prod_{i=0}^{t-1} \Pr_{LM}(\mathbf{r}_{i+1}|\mathbf{q}, \mathbf{r}_{[i]}). \tag{2}$$

As Equation 2 suggested, at timestamp $i$, the language model generates the next reasoning step $\mathbf{r}_i$ based on previous premises, which is $\mathbf{r}_i \sim \Pr_{LM}(\mathbf{r}_i|\mathbf{q}, \mathbf{r}_{[i-1]})$. This formulation follows the principle of deductive reasoning.

### 2.2   Step-wise Beam Search

Under beam size $m$, traditional beam search decodes at token level, which stores Top-$m$ candidate tokens, and uses them for future decoding. Formally, we denote the log-probability of LM generating the $k$-th tokens as $\phi(y_k) = \log \Pr_{LM}(y_k|y_1, y_2, ..., y_{k-1}, \mathbf{x}) = \log \Pr_{LM}(y_k|\mathbf{y}_{[k-1]}, \mathbf{x})$, and the log-probability of a solution at timestamp $k$ as $\Phi(\mathbf{y}_{[k]}) = \sum_{i \in [k]} \phi(y_i)$. Given a set of $m$ previous solutions at timestamp $i$ as $Y_{i-1} = \{\mathbf{y}_{[i-1]}^1, \mathbf{y}_{[i-1]}^2, ..., \mathbf{y}_{[i-1]}^m\}$, beam search generates as:

$$Y_{[i]} = \underset{\mathbf{y}_{[i]}^1, \mathbf{y}_{[i]}^2, ..., \mathbf{y}_{[i]}^m}{\arg\max} \sum_{k \in [m]} \Phi(\mathbf{y}_{[i]}^k). \tag{3}$$

However, in reasoning tasks, it is hard to verify whether a single token is deducible. Thus, we assign a reasoning step $\mathbf{r}$ as the minimal unit in step-wise beam search. Formally, we denote the log-probability of generating the $k$-th reasoning step as $\psi(\mathbf{r}_k) = \Phi(\mathbf{y})$ and the log-probability of a solution at timestamp $k$ to be $\Psi(\mathbf{r}_{[k]}) = \sum_{i \in [k]} \psi(\mathbf{r}_i)$. Given a set of $m$ previous solutions at timestamp $i$ as $R_{[i-1]} = \{\mathbf{r}_{[i-1]}^1, \mathbf{r}_{[i-1]}^2, ..., \mathbf{r}_{[i-1]}^m\}$, step-wise beam search infers as:

$$R_{[i]} = \underset{\mathbf{r}_{[i]}^1, \mathbf{r}_{[i]}^2, ..., \mathbf{r}_{[i]}^m}{\arg\max} \sum_{k \in [m]} \Psi(\mathbf{r}_{[i]}^k). \tag{4}$$

Combining multi-step CoT reasoning with step-wise beam search balances exploration and exploitation in reasoning tasks. However, confidence scores from language models cannot verify logical consequence between a reasoning step and its premises. To tackle this problem, we propose to constrain the step-wise beam search with deductive scores.

### 2.3 Deductive Verification Constrained Beam Search

To verify the logical coherence between a reasoning step and its premises, we propose to train a deductive verifier since LLM itself often fails to detect reasoning errors (Hong et al., 2023). Formally, given premises $\mathbf{c}_{[i]} = [\mathbf{c}_1, \mathbf{c}_2, ..., \mathbf{c}_i]$ and the candidate reasoning step $\mathbf{r}$, the deductive score can be formulated as: $s = f(\mathbf{c}_{[i]}, \mathbf{r}) = \Pr_f(\mathbf{r}|\mathbf{c}_1, \mathbf{c}_2, ..., \mathbf{c}_i)$, where $f$ is the deductive verifier function. The details of the deductive verifier are illustrated in Section 3. Then, we utilize the deductive verifier to constrain the step-wise beam search. To clearly illustrate the process, we show the case how, given one antecedent solution beam $\mathbf{r}_{[i-1]} \in R_{[i-1]}$ at timestamp $i$, reasoning steps are sampled and scored.

In the exploration phase of beam search, the language model samples a list of potential reasoning steps. Concretely, for sampling times $n$, the question $\mathbf{q}$ and $\mathbf{r}_{[i-1]}$ form the current context $\mathbf{c}_{[i]} = [\mathbf{q}, \mathbf{r}_{[i-1]}]$, and we can sample a set of $n$ possible reasoning steps $\hat{R}_i = \{\mathbf{r}_1, \mathbf{r}_2, ..., \mathbf{r}_n\}$, where $\mathbf{r} \sim \Pr_{LM}(\mathbf{r}|\mathbf{q}, \mathbf{r}_{[i-1]})$. Concatenating $\mathbf{r} \in \hat{R}_i$ with $\mathbf{r}_{[i-1]}$ generates candidate reasoning chains set $\hat{R}_{[i]} = \{[\mathbf{r}_{[i-1]}, \mathbf{r}_1], [\mathbf{r}_{[i-1]}, \mathbf{r}_2], ..., [\mathbf{r}_{[i-1]}, \mathbf{r}_n]\}$ at timestamp $i$.

In terms of exploitation, instead of using the language model probability $\Pr_{LM}(\mathbf{r}|\mathbf{c}_{[i]})$ to evaluate these reasoning steps, deductive verification scores $S = \{s_1, s_2, ..., s_n\}$ of candidate reasoning paths $\hat{R}_i$ are applied. Each score $s_j, j \in [n]$ is calculated by multiplying the score of $\mathbf{r}_{[i-1]}$ and the score of each candidate reasoning step, that is:

$$s_j = \mathbf{s}([\mathbf{r}_{[i-1]}, \mathbf{r}_j]) = \mathbf{s}(\mathbf{r}_{[i-1]}) \times \Pr{}_f(\mathbf{r}_j|\mathbf{q}, \mathbf{r}_{[i-1]}) = \prod_{k=1}^{i} \Pr{}_f(\mathbf{r}_k|\mathbf{q}, \mathbf{r}_{[k-1]}), \quad (5)$$

which follows the autoregressive factorization form and allows us to apply on the beam search algorithm.

Consequently, LM generates $n$ times for each beam at each step, sampling a total number of $m \times n$ candidate reasoning steps. After scoring on these steps, the top $m$ of them are selected according to the deductive score. This cycle of exploration and exploitation repeats until the final answer is generated or it reaches the upper limit of reasoning length.

## 3 Deductive Verifier

As stated above, a deductive verifier evaluates whether the reasoning step can be deduced from previous contexts, which resembles a natural language inference (NLI) task. Thus, we use *deberta-v3-large* (He et al., 2021), which achieves the best performance across various NLI benchmarks despite its small size, as the backbone. A small scalar head is adopted to predict deductive scores based on embedding the [CLS] token.

However, the difficulties of training a deductive verifier primarily reside in the training data quality and the training method. Changing one single token could lead to various errors, which is hard for any model to detect. Furthermore, the lack of high-quality false deductive reasoning step hinges the training of the verifier. To fully understand how LLMs make mistakes, we dive into the incorrect samples generated by LLMs. From the perspective of deductive reasoning, there are two main classes of reasoning errors: grounding errors and logic errors (Ling et al., 2024). Most grounding errors happening in the reasoning process can be detected by finding the contradiction between the context and the rationales, while the latter ones are illogical reasoning steps deduced from the previous context.

Thus, we propose a scalable and labor-free data construction method and a ranking-based training framework to teach the verifier to detect false reasoning steps. The whole training is divided into two stages. In stage 1, we heuristically corrupt gold reasoning steps to

| Context | Type | Not. | Reasoning Step | Error Reason |
|---|---|---|---|---|
| Randy has some money. He spent $10 buying his lunch. He spent a quarter of the money he had left on an ice cream cone. If the ice cream cone cost $5, what is the amount of money, in dollars, Randy had at first? | Gold | $\mathbf{r}$ | Randy has 5*4=20 dollars left after buying lunch. | - |
| | Grounding | $\mathbf{r}'_1$ | Randy has 10*4=40 dollars left after buying lunch. | Minor token-level error, hard for models to detect. |
| | Logic | $\mathbf{r}'_2$ | At first, Randy had a sum of 20+10=30 dollars. | Logic-level error caused by reversed steps, not following deductive reasoning. |
| | Irrelevant | $\mathbf{r}'_3$ | He eats 65 black cookies from the cookie jar, with 1/2 * 130 = 65. | Major error, completely incoherent with the context. |

Table 1: Examples of heuristically synthesized false reasoning steps.

simulate typical false reasoning and train the verifier to detect them. In stage 2, the verifier trained from stage 1 is deployed to detect potential false reasoning steps generated by LLMs, bridging the gap between synthetic data and real-world data. Consequently, the model from stage 1 is continue-trained.

## 3.1 A General Deductive Verifier

In the first stage, we require the verifier to detect two general types of reasoning errors: grounding error and logic error. However, such fine-grained step-wise data is hard to annotate. Thus, we propose to synthesize false reasoning steps automatically.

Since it is hard to edit natural language to meet our demands, we turn to arithmetic reasoning, which can be viewed as a middle ground between symbols and natural language. In terms of reasoning steps with grounding errors, we randomly replace one of the numbers on the left side of the equation in the gold reasoning step with numbers existing in previous contexts or randomly generated numbers to simulate false grounding or hallucinations. As for logic errors, we randomly select reasoning steps after the current gold reasoning step. Under this circumstance, the reasoning process is reversed and disrupted, making it a logic error. Moreover, to enhance the understanding of the model for this task, we use randomly selected reasoning steps across the whole dataset as an irrelevant false reasoning step. The examples of these errors are shown in Table 1.

To provide fine-grained supervision for error detection, we use margin ranking (Shashua & Levin, 2002) to model the task. Specifically, given context $\mathbf{c}$, gold reasoning step $\mathbf{r}$, and three false reasoning steps $\mathbf{r}'_1$, $\mathbf{r}'_2$, and $\mathbf{r}'_3$, respectively representing grounding error, logic error, and irrelevant reasoning step, the verifier $f$ scores all the candidates through $s = f(\mathbf{c}, \mathbf{r})$, which outputs four scores $s$, $s'_1$, $s'_2$, and $s'_3$. Then, the loss of ranking these reasoning steps is formulated as the weighted sum of three margin ranking losses:

$$\mathcal{L} = -\sum_{i=1}^{3} \alpha_i \times (s - s'_i - m_i), \tag{6}$$

where $m_i$ is the hyper-parameter controlling the margin and $\alpha_i$ weighs each loss.

## 3.2 Deductive Verifier with Model Feedback

In the first stage, we train a general deductive verifier, but the wrong samples synthesized heuristically are less diverse than the ones encountered during inference. To bridge the gap between synthesized data and real-world data, we use the verifier from stage 1 to detect false reasoning steps generated by an actual language model, where we choose *Llama2-7b* for the generation. The reason why we choose a relatively small language model for the generation is to maximize the diversity and the likelihood of generating incorrect steps.

To be concrete, given the verifier $f_1$ trained by stage 1, we feed context $\mathbf{c}$ into the LLM and sample 10 reasoning steps. Then, these steps are scored and ranked by $f_1$. From this ranking, we select the reasoning step that exhibits the most significant decrease in the deductive score, designating it as the hard negative sample. We replace $\mathbf{r}'_1$ with the generated hard negative sample as $\mathbf{r}''_1$, keeping the original way of generating $\mathbf{r}'_2$ and $\mathbf{r}'_3$. Consequently, we continue training the verifier $f_1$ by Equation 6 with a smaller learning rate.

| Method | Arithmetic Reasoning | | | | | | Commonsense Reasoning | | | Symbolic Reasoning | |
|---|---|---|---|---|---|---|---|---|---|---|---|
| | GSM8K | SVAMP | AQuA | SingleEq | MultiArith | Avg.↑↓ | StrategyQA | CSQA | Avg.↑↓ | Coin | Avg.↑↓ |
| *Llama2-7b* | | | | | | | | | | | |
| Greedy | 22.0 | 49.0 | 3.2 | 67.5 | 68.3 | | 64.0 | 66.9 | | 53.8 | |
| **DBS** | 31.2 | 55.0 | 5.7 | 69.0 | 74.4 | +5.3 | 66.4 | 67.0 | +2.6 | 51.6 | -2.2 |
| SC | 28.1 | 56.7 | 4.9 | 77.5 | 77.8 | | 65.6 | 67.2 | | 53.0 | |
| **DBS + SC** | 32.1 | 59.3 | 8.5 | 78.9 | 85.6 | +3.9 | 67.6 | 68.3 | +1.6 | 54.1 | +1.1 |
| *Llama2-13b* | | | | | | | | | | | |
| Greedy | 35.6 | 52.3 | 2.8 | 72.2 | 70.6 | | 66.2 | 53.2 | | 60.2 | |
| **DBS** | 43.2 | 58.0 | 6.1 | 76.7 | 85.6 | +7.2 | 64.6 | 53.7 | -0.6 | 61.2 | +1.0 |
| SC | 42.0 | 68.3 | 3.6 | 86.4 | 91.7 | | 65.4 | 68.0 | | 61.8 | |
| **DBS + SC** | 45.2 | 72.0 | 9.3 | 90.7 | 94.4 | +3.9 | 66.6 | 69.8 | +1.5 | 63.4 | +1.6 |
| *Llama2-70b* | | | | | | | | | | | |
| Greedy | 41.7 | 51.3 | 10.1 | 70.0 | 70.6 | | 69.8 | 59.4 | | 71.2 | |
| **DBS** | 58.3 | 61.7 | 10.1 | 78.9 | 90.6 | +11.2 | 70.6 | 62.4 | +1.9 | **80.4** | +8.7 |
| SC | 64.8 | 79.3 | 10.5 | 91.3 | 97.2 | | 74.0 | 74.0 | | 79.6 | |
| **DBS + SC** | 67.6 | 79.3 | 14.5 | 92.7 | 97.2 | +1.6 | **75.0** | 73.3 | +0.2 | 80.2 | +0.6 |
| *ChatGPT* | | | | | | | | | | | |
| Greedy | 68.8 | 72.0 | 16.5 | 95.1 | 97.2 | | 65.4 | 65.1 | | 75.1 | |
| **DBS** | 75.9 | 75.7 | 24.8 | 92.8 | 97.8 | +3.2 | 68.6 | 74.0 | +6.2 | 75.5 | +0.4 |
| SC | 81.3 | 81.3 | 20.2 | **97.6** | 98.3 | | 70.6 | 75.4 | | 78.9 | |
| **DBS + SC** | **83.5** | **82.7** | **28.8** | 97.0 | **99.4** | +2.5 | 69.8 | **78.3** | +1.1 | 79.5 | +0.6 |

Table 2: The result comparison on arithmetic reasoning, commonsense reasoning, and symbolic reasoning tasks. The results represent accuracy (%) on each dataset. **Bold** indicates best results and underline indicates second bests.

# 4 Experimental Setup

## 4.1 Reasoning Tasks

For our evaluation, we choose benchmarks from 3 different reasoning genres, namely, arithmetic reasoning, commonsense reasoning, and symbolic reasoning. These 3 types of reasoning tasks represent diverse reasoning paradigms.

**Arithmetic Reasoning.** Following Li et al. (2023) and Ling et al. (2024), we choose GSM8K (Cobbe et al., 2021), SVAMP (Patel et al., 2021), AQuA (Ling et al., 2017), SingleEq (Koncel-Kedziorski et al., 2015), and MultiArith (Roy & Roth, 2016) for evaluation. For AQuA, we evaluate the accuracy by comparing with the answer of the ground truth.

**Commonsense Reasoning.** Following Li et al. (2023), we use CommonsenseQA (Talmor et al., 2019) and StrategyQA (Geva et al., 2021). CommonsenseQA asks the model to choose the best answer from 5 choices, and StrategyQA asks for a True/False answer.

**Symbolic Reasoning.** We use the Coin Flip dataset (Wei et al., 2022b). The task is to determine which face of the coin is up after a series of operations.

## 4.2 Details

**Language Models.** We evaluate our method on models of various scales, including *Llama2-7b*, *Llama2-13b*, *Llama2-70b* (Touvron et al., 2023), and ChatGPT (*gpt-3.5-turbo-instruct*) (OpenAI, 2022). These models represent different levels of reasoning abilities. For the verifier, we choose *deberta-v3-large* as the backbone of our verifier. The training details are in Appendix B.

**Prompts.** For arithmetic reasoning tasks, we apply one prompt to all tasks. For commonsense reasoning tasks and the symbolic reasoning task, we write a prompt for each task to ensure the model can output the correct answer format. All methods are evaluated by the same prompt on each task. The details of prompts are in Appendix B.3.

**Baselines.** To prove the effectiveness of DBS, we compare with greedy decoding algorithm (Jurafsky & Martin) and self-consistency (Wang et al., 2022). For other SOTA baselines, we choose SelfEval (Xie et al., 2023) and Deductive Verification (Ling et al., 2024), which do not update the parameters of LLMs. The former represents SOTA decoding algorithm, while the latter stands for methods utilizing novel procedure design to conduct deductive reasoning. Since a full-scale experiment requires excessive token cost due to the extensive search and verification of these methods, we provide results on the GSM8K dataset.

**Inference.** During inference, we set beam size $m$ to 5 and sampling times $n$ to 10. For all models and baselines, we use their default parameter settings for generation.

# 5 Main Result

Table 2 demonstrates the overall performance of the methods. We compare DBS with baselines under two paradigms: single chain setting and multiple chain setting. In multiple chain setting, the generated outcomes are integrated with self-consistency. Table 3 presents a comparative analysis of our approach against SOTA baselines.

## 5.1 Effectiveness

As shown in Table 2, DBS improves the performance across models of different scales and diverse reasoning tasks. For the single chain setting, the improvement is substantial. On arithmetic reasoning tasks, taking GSM8K as an example, we observe an increase from 7.6% to 16.6% across models of various scales. Specifically, with *Llama2-7b* and *gpt-3.5-turbo-instruct*, DBS yields improvements of 9.2% and 7.0%, respectively, affirming the effectiveness of our proposed strategy. On commonsense reasoning tasks and symbolic reasoning tasks, we can expect an average increase of 2.5%/2.0% on models of all scales.

Regarding the multiple reasoning chain setting, DBS outperforms naive self-consistency. Concretely, we can see an average of 3.0% improvement on arithmetic reasoning tasks, 0.5% on commonsense reasoning tasks, and 1.0% on symbolic reasoning tasks, respectively. On the SingleEq and StrategyQA datasets, the performance of DBS on ChatGPT is slightly lower (-0.6%/-0.8%). These datasets require fewer reasoning steps for the final answer, as opposed to our paradigm of multiple reasoning steps. Nevertheless, the universal improvements demonstrate the effectiveness of our proposed method.

## 5.2 Comparison with Current Solutions

Table 3 compares our decoding strategy with previous SOTA reasoning strategies. The comparative results, grounded in accuracy and token cost metrics, substantiate our approach's effectiveness and token efficiency. Notably, the self-evaluate pattern performs worse when the scale of the LLM drops. Moreover, it consumes an excessive amount of tokens during evaluation. In contrast, DBS enhances the performance by approximately 10% on *Llama2-7b* and performs better than the baselines on *gpt-3.5-turbo-instruct* across the paradigms of single

| Model | Method | Accu | Tokens | #Rationales |
|---|---|---|---|---|
| Llama2-7b | SelfEval | 21.8 | 50M | 1 |
| | SelfEval + SC | 24.2 | | 10 |
| | DV | 10.5 | 372M | 1 |
| | DV + SC | 13.2 | | 10 |
| | **DBS** | 31.2 | 4M | 1 |
| | **DBS + SC** | **32.1** | | 10 |
| ChatGPT | SelfEval | 71.3 | 12M | 1 |
| | SelfEval + SC | 74.7 | | 10 |
| | DV | 68.2 | 409M | 1 |
| | DV + SC | 83.3 | | 10 |
| | **DBS** | 75.9 | 5M | 1 |
| | **DBS + SC** | **83.5** | | 10 |

Table 3: Comparison of between DBS and different reasoning methods.

and multiple reasoning chains. Furthermore, it consumes much fewer input and output tokens. Consuming 80 times less tokens, DBS outperforms DV on *gpt-3.5-turbo-instruct*.

# 6 Analysis

We conduct a detailed analysis to investigate the verifiability and efficiency of our proposed method. Moreover, we show how our method can adapt to different settings.

## 6.1 Verifier Analysis

We comprehensively analyze the verifiability of our proposed verifier on performance (Table 4) and score distribution (Figure 3).

| Method | MRR | HITS@1 | HITS@3 | HITS@5 |
|---|---|---|---|---|
| ChatGPT | 0.48 | 0.32 | 0.49 | 0.67 |
| Our Verifier | **0.59** | **0.41** | **0.69** | **0.86** |

Table 4: Verification ability comparison of our verifier and ChatGPT.

**Empirical Study.** For empirical experiments, we test how our verifier will score the gold reasoning steps compared to synthesized reasoning steps. We randomly sample 500 context-step pairs from the test set of the

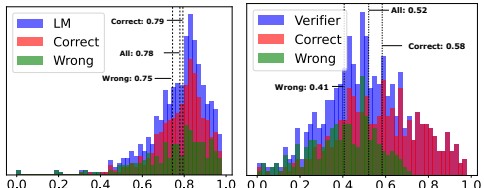

(a) LM distribution     (b) Verifier distribution

Figure 3: Distributions of language model and verifier scores on reasoning paths.

| Method | Accuracy | Tokens |
|---|---|---|
| Greedy | 21.99 | 267,462 |
| Self-consistency, $n = 10$ | 28.05 | 1,984,921 |
| **DBS**, $m = 1, n = 10$ | 29.34 | 1,414,435 |
| **DBS**, $m = 5, n = 10$ | **31.16** | 4,042,053 |

Table 5: Cost Analysis of our method and other baseline methods. $m$ represents beam size and $n$ represents sampling times.

GSM8K dataset as gold reasoning steps. For each pair, we synthesize nine reasoning steps using *Llama2-7b* as inferior reasoning steps. The verifier's task is to rank them, and the performance is evaluated by four metrics, namely, Mean Reciprocal Rank (MRR), HITS@1/3/5. MRR evaluates the average rank of the gold reasoning step, and the HITS metrics reflect whether the gold reasoning step will be chosen under beam size setting 1/3/5. To compare with the self-evaluation pattern, we ask ChatGPT to rank these reasoning steps rather than predict scores, leveraging its inherent reranking capabilities (Ma et al., 2023). The results are listed in Table 4. Our verifier outperforms ChatGPT across all metrics, evidencing its capability. Notably, our verifier correctly identifies 86% of the gold reasoning steps within the top 5 positions out of 10 samples, affirming the deducibility of the reasoning paths decoded under $m = 5, n = 10$.

**Distribution Analysis.** To ascertain the reliability of our verifier, we compare the score distributions for correct and wrong predictions between the original LM confidence (*gpt-3.5-turbo-instruct*) and our deductive verification score. We use results from greedy decoding, which naturally produces confidence scores from LM, and ask the verifier to score on them. Figure 3 shows the substantial difference between an LM confidence score and our deductive score. Notably, the LM confidence score demonstrates a mere 4% increase in scores of the correct reasoning paths, whereas our verifier exhibits a 17% increase. This significant difference proves the enhanced verifiability of our verification approach.

## 6.2 Cost Analysis

The cost of sampling multiple times is enormous. We analyze the cost of our methods under different settings and compare them with the baselines. Specifically, we compare our approach with greedy decoding, self-consistency, SelfEval (Xie et al., 2023), and DV (Ling et al., 2024). The results are presented in Table 3 and Table 5. Our analysis reveals that greedy decoding is the most token-economic, as it does not involve any form of sampling, but its performance lags. When $m$ is constrained to 1, the token generation is minimized even further than that required by self-consistency strategies. Still, our method demonstrates higher accuracy. Moreover, DBS proves more effective and token-efficient under the same beam size than those leveraging LLMs' self-evaluation capabilities.

## 6.3 Commonsense Reasoning Task

Unlike arithmetic or symbolic reasoning tasks, commonsense reasoning tasks leverage the model's intrinsic commonsense. Thus, we introduce a prompt strategy, which asks the model to recall the necessary commonsense via greedy decoding, followed by DBS navigating the reasoning over the elicited contexts. We compare this setting to standard settings and baselines, and the results are shown in Table 6. The performance varies from different tasks, at-

| Method | StrategyQA | CSQA |
|---|---|---|
| Greedy | 64.00 | 66.91 |
| **DBS** | 66.40 | 66.99 |
| *- w. recall prompt* | 65.40 | 66.67 |
| SC | 65.60 | 67.24 |
| **DBS** | **67.60** | 68.29 |
| *- w. recall prompt* | 65.80 | **69.45** |

Table 6: Different prompt settings on commonsense reasoning tasks. We use *Llama2-7b* as the backbone for all methods.

tributed to the distinct nature of the tasks. Concretely, StrategyQA typically requires a 2-3 step knowledge recall followed by a single reasoning step, whereas CSQA demands a 3-5

| Question | Greedy | Ours |
|---|---|---|
| James decides to run 3 sprints 3 times a week. He runs 60 meters each sprint. How many total meters does he run a week? | He runs 3 times a week, which is equal to 3 x 7 = 21 sprints. `0.0102` The distance he runs each sprint is 60 meters, so the total distance is 60 x 21 = 1320 meters. `0.4963` Final Answer:1320 | Each sprint is 60 meters. `0.8270` So he runs 60 x 3 = 180 meters each day. `0.9643` He runs each sprint 3 times a week. `0.9983` Therefore, he runs a total of 180 x 3 = 540 meters per week. `0.9463` Final Answer:540 |
| Toulouse has twice as many sheep as Charleston. Charleston has 4 times as many sheep as Seattle. How many sheep do Toulouse, Charleston, and Seattle have together if Seattle has 20 sheep? | Toulouse has twice as many sheep as Charleston, so Toulouse has 2 x 4 = 8 sheep. `0.3149` Charleston has 4 times as many sheep as Seattle, so Charleston has 4 x 20 = 80 sheep. `0.9877` So Toulouse has 8 + 80 = 88 sheep. `0.3923` Seattle has 20 sheep. `0.7479` Final Answer:88 + 20 + 88 = 196 | Charleston has 4 times as many sheep as Seattle. Charleston has 20 x 4 = 80 sheep. `0.9912` Toulouse has twice as many sheep as Charleston. Charleston has 80 sheep. Toulouse has 80 x 2 = 160 sheep. `0.9736` Finally, add the number of sheep Toulouse, Charleston, and Seattle have. `0.9981` So the total number of sheep they have together is 160 + 80 + 20 = 260 sheep. `0.9924` Final Answer:260 |

Table 7: Two GSM8K examples from *Llama2-7b* scored with our verifier.

step recall process alongside multiple reasoning steps. Our findings suggest that the recall prompt is more suitable for tasks demanding multi-step reasoning.

## 6.4 Case Study

Table 7 presents two GSM8K examples from *Llama2-7b*. The first example demonstrates a scenario where hallucination emerges under greedy decoding, which our verifier identifies with an extremely low score, thereby precluding its selection by our deductive decoding strategy. On the contrary, every reasoning step from our reasoning path is deduced from the previous context and is scored much higher than the incorrect steps. In the second example, a grounding error occurs in steps marked by red scores. Although the wrong reasoning steps resemble the correct ones, our verifier detects these minor errors. In both examples, the reasoning path generated by our decoding strategy initiates the reasoning by listing premises, followed by one reasoning step. This pattern strictly follows the principle of deductive reasoning, making the generated results more deducible.

## 7 Related Work

**Answer Aggregation.** Sampling techniques of language models, such as temperature sampling (Ackley et al., 1985), top-k sampling (Fan et al., 2018), and top-p sampling (Holtzman et al., 2019), bring diversity to the outcome but also uncertainty to the reasoning process, which is not favored in reasoning tasks. These methods aim to reduce uncertainty in the reasoning process by aggregating answers from sampled reasoning paths. After sampling diverse outputs from LLMs, Wang et al. (2022) propose to use consistency as the metric to aggregate the answers. Other methods evaluate whether the reasoning step can lead to the correct answer by training a verifier (Li et al., 2023; Wang et al., 2023).

**Self-Evaluation.** Recent research on reducing reasoning errors is inclined to follow the self-verify-then-correct pattern (Dhuliawala et al., 2023; Weng et al., 2022; Zhang et al., 2023b; Ling et al., 2024; Miao et al., 2023). They design different procedures and prompts to achieve better performance. Taking two typical approaches as examples, Dhuliawala et al. (2023) design a chain-of-verification procedure to verify facts from its outputs, and Miao et al. (2023) ask LLM to detect errors in their step-by-step rationales. However, recent works (Huang et al., 2023; Hong et al., 2023) have pinpointed a critical limitation of LLMs in self-correction during reasoning tasks. Their findings suggest that LLMs indiscriminately alter reasoning steps without external feedback, irrespective of their initial accuracy.

**Decoding Strategies.** Conventional decoding strategies include greedy decoding (Teller, 2000), which selects tokens with the highest probabilities, and beam search (Graves, 2012), which stores candidate beams for future prediction. In the era of LLMs, these decoding strategies are implemented at a more coarse-grained level, especially on reasoning tasks involving multiple steps. They decompose the reasoning process into steps (Khot et al., 2022) and apply decoding or search algorithms (Yao et al., 2023; Xie et al., 2023).

## 8 Conclusions

In this paper, we aim to eliminate errors in intermediate reasoning steps in CoT reasoning, making it more reliable. To this end, we propose **Deductive Beam Search** that integrates CoT with step-wise beam search and scores each reasoning step with a deductive verifier, which verifies whether the reasoning step is a logical consequence. Beam search explores by sampling potential reasoning steps, while the verifier exploits by selecting the most deducible steps. To train such a verifier, we propose a scalable and labor-free data construction method. It initiates by heuristically introducing errors into gold reasoning steps and enhances the diversity and difficulty of training data by synthesizing hard negatives through the verifier trained on those typical wrong steps. Extensive experiments show our method's effectiveness across various model scales and diverse reasoning tasks without changing the standard CoT paradigm and parameters of LLMs. Further analysis proves the verifiability and robustness endowed by our verifier, thereby significantly improving the deducibility of the generated reasoning paths.

**Acknowledgments**

The authors would like to thank Jiahao Shi and Yikai Zhang from Fudan University, and Yifei Li from the Ohio State University, as well as the anonymous reviewers for their valuable comments.

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

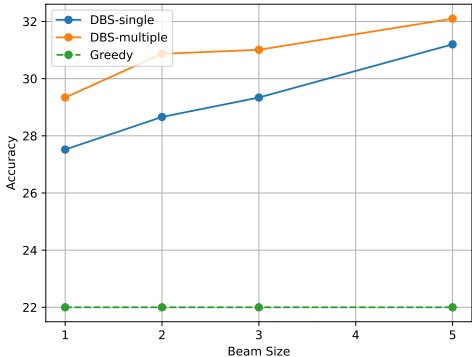

Figure 4: Accuracy under different beam sizes on different models.

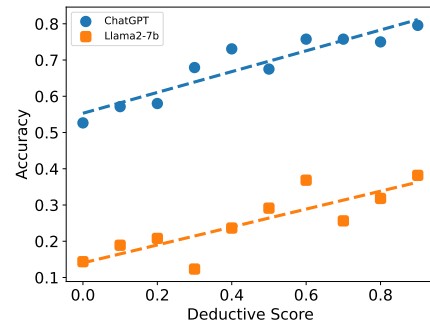

Figure 5: Accuracy under different deductive score thresholds on greedy decoding results.

## A    Extended Experiments

**Ablation study on beam size.**   We conduct experiments on how beam size affects performance. Figure 4 shows the trend of DBS performance under single chain setting and multiple chain setting. The performance steadily grows when the beam size rises.

**Verifier Robustness.**   To ensure that our verifier can equally verify reasoning steps generated by different models, we visualize the accuracy under different deductive score thresholds for *Llama2-7b* and ChatGPT, as depicted in Figure 5. The lines in the figure represent polynomial fits of the data. Their near-parallel alignment suggests the robustness of performance improvement across these models as the threshold increases. Intriguingly, these lines also offer insights into the inductive reasoning capabilities of the two models.

Ideally, the accuracy approaches zero at the deductive score zero. However, the observed non-zero accuracy suggests the models' inductive reasoning capabilities. Although the reasoning process might not align with the deductive reasoning paradigm, LMs can still arrive at correct conclusions, likely by intuitively skipping over specific reasoning steps, which is the act of inductive reasoning.

**DBS Robustness.** To demonstrate that DBS can sustain its accuracy as the length of the reasoning steps increases, we conducted experiments to analyze the impact of reasoning step length on the final outcome's ac-

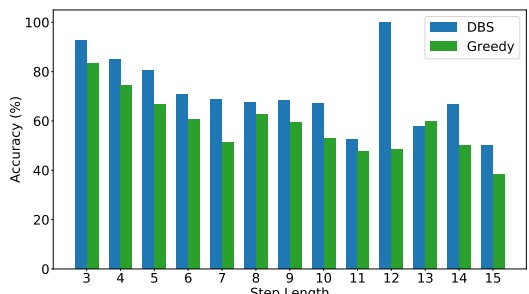

Figure 6: Accuracy under different reasoning step length.

curacy. The results, presented in Figure 6, indicate that DBS consistently achieves higher accuracy, even when the reasoning process extends to 15 steps.

## B    Experimental Details

### B.1    Training Data

At stage 1, we choose GSM8K dataset to train the general deductive verifier. The gold rationales provided are decomposed into sentences as gold reasoning steps. After filtering out some steps that cannot be altered into false reasoning steps, we construct a training dataset of 22,362 samples. At stage 2, the verifier from stage 1 is used to generate hard negative reasoning steps as stated in Sec. 3. We choose *Llama2-7b* as our language model to generate candidate false reasoning steps. For arithmetic reasoning and symbolic reasoning tasks, we

---

Given the question, please give the rationales step by step and give a final answer.

Example 1:
Question:
Kate's hair is half as long as Emily's hair.
Emily's hair is 6 inches longer than Logan's hair.
If Logan hair is 20 inches, how many inches is Kate's hair?
Answer:
Emily's hair is 20-6 = 14 inches long.
Kate's hair 14/2= 7 inches long.
Final Answer:7

Example 2:
Question:
John puts $25 in his piggy bank every month for 2 years to save up for a vacation.
He had to spend $400 from his piggy bank savings last week to repair his car.
How many dollars are left in his piggy bank?
Answer:
He saved money for 2 years, which is equal to 12 x 2 = 24 months.
The amount of money he saved is $25*24 = $600.
But he spent some money so there is $600 - $400 = 200 left.
Final Answer:200

---

Table 8: Prompt for arithmetic reasoning tasks.

use the MetaMathQA dataset to generate training data. For commonsense reasoning tasks, we use the StrategyQA dataset to generate training data. Finally, we train the arithmetic verifier on 150,000 samples and the commonsense verifier on 5,000 samples.

## B.2  Training Details

At stage 1, we finetune *deberta-v3-large* with learning rate $1 \times 10^{-5}$ and batch size 128. As for margins, we set the margins between the gold reasoning step and grounding error step/logic error step/irrelevant step to 0.3/0.6/0.9. At stage 2, we continue to finetune the verifier from stage 1 with learning rate $1 \times 10^{-6}$ and batch size 128. As for margins, we set the margins between the gold reasoning step and hard negative reasoning step/logic error step/irrelevant step to 0.3/0.6/0.9.

## B.3  Prompts

For the results in Table 2, we use the following prompts:

- Arithmetic reasoning tasks share the same prompt, shown in Table 8.
- For StrategyQA, we use the prompt in Table 9.
- For CSQA, we use the prompt in Table 10.
- For Coin, we use the prompt in Table 11.

For the results in Table 6, we use the following prompts:

- For StrategyQA, we use prompt in Table 12.
- For CSQA, we use prompt in Table 13.

---

Given the question, output the rationale step by step and give the final answer (yes or no).

Example 1
Question:
Do hamsters provide food for any animals?
Answer:
Hamsters are prey animals.
Prey are food for predators.
Final answer: yes

Example 2
Question:
Could a llama birth twice during War in Vietnam (1945-46)?
Answer:
The War in Vietnam was 6 months.
The gestation period for a llama is 11 months, which is more than 6 months.
Final answer: no

---

Table 9: Prompt for StrategyQA.

---

Given the question, output the rationale step by step and give the final answer. You should choose the best answer.

Example 1
Question:
Sammy wanted to go to where the people were. Where might he go?
A. race track
B. populated area
C. the desert
D. apartment
E. roadblock
Answer:
Sammy wanted to go to places with many people.
Race track and apartment do not have many people.
The desert and roadblock have few people.
And, the populated area means that it is the place with many people.
Thus, Sammy should go to populated area.
Final Answer: B

Example 2
Question:
The fox walked from the city into the forest, what was it looking for?
A. pretty flowers
B. hen house
C. natural habitat
D. storybook
E. dense forest
Answer:
The forest does not have hen house or storybook.
The fox is a carnivore that does not look for flowers and forest.
The forest is a natural habitat for foxes.
Thus, it was looking for a natural habitat.
Final Answer: C

---

Table 10: Prompt for CSQA.

Given the question, output the rationale step by step and give the final answer.

Example 1
Question:
A coin is heads up. sager does not flip the coin. zyheir flips the coin.
Is the coin still heads up?
Answer:
sager does not flip the coin, so the coin is heads up.
zyheir flips the coins, so the coin is tails up.
Final Answer: no

Example 2
Question:
A coin is heads up. mailey does not flip the coin. maurisa does not flip the coin.
Is the coin still heads up?
Answer:
mailye does not flip the coin, so the coin is heads up.
maurisa does not flip the coin, so the coin is heads up.
Final Answer: yes

Table 11: Prompt for Coin.

Given the question, output the rationale step by step and give the final answer (yes or no).

Example 1
Question:
Do hamsters provide food for any animals?
Answer:
Fact:
Hamsters are prey animals.
Prey are food for predators.
Reasoning:
Hamsters are food for some predators.
Final answer: yes

Example 2
Question:
Could a llama birth twice during War in Vietnam (1945-46)?
Answer:
Fact:
The War in Vietnam was 6 months.
The gestation period for a llama is 11 months, which is more than 6 months.
Reasoning:
A llama could not birth twice during War in Vietnam.
Final answer: no

Table 12: Prompt for StrategyQA with prompt.

Given the question, output the rationale step by step and give the final answer.
You should choose the best answer.

Example 1
Question:
Sammy wanted to go to where the people were. Where might he go?
A. race track
B. populated area
C. the desert
D. apartment
E. roadblock
Answer:
Fact:
Sammy wanted to go to places with many people.
Race track and apartment do not have many people.
The desert and roadblock have few people.
And, the populated area means that it is the place with many people.
Reasoning:
Thus, Sammy should go to populated area.
Final Answer: B

Example 2
Question:
The fox walked from the city into the forest, what was it looking for?
A. pretty flowers
B. hen house
C. natural habitat
D. storybook
E. dense forest
Answer:
Fact:
The forest does not have hen house or storybook.
The fox is a carnivore that does not look for flowers and forest.
The forest is a natural habitat for foxes.
Reasoning:
Thus, it was looking for a natural habitat.
Final Answer: C

Table 13: Prompt for CSQA with recalling commonsense first.

