# OpenReview forum: "Deductive Beam Search: Decoding Deducible Rationale for Chain-of-Thought Reasoning"
_colmweb.org/COLM/2024/Conference — COLM_

### Official Review · Reviewer_R7sC · 2024-05-08

**Rating:** 4
**Confidence:** 5
**Ethics Flag:** 1

**Summary:**

This paper propose a new prompting method to improve the reasoning capability of llms. Specifically, instead of using greedy search, this paper proposed to conduce beam search based on the step level with the help of an additional NLI model to provide the beam ranking signal. Experiments on several datasets show that this method can achieve marginal improvement over the majority voting baseline.

**Questions To Authors:**

1. What is the generalization capability of the proposed verifier. Can you test on a more general benchmark such as big-bench hard?
2. Can you elaborate more on the implementation of the self-consistency baseline including all the important hyper-parameters?

**Reasons To Accept:**

This paper combines a sound search algorithm (i.e., beam search) and the CoT process to improve the model's complex reasoning capability.

**Reasons To Reject:**

1. Combining advanced search algorithms into the reasoning process of large language models is a hot research topic in the community. There are several existing works that applies beam search, tree search, or MCTS into the reasoning process. This paper is missing the comparison of these baselines. I could think of a key difference between this work and other baselines is that the proposed deductive verifier is task-independent while other methods either use signal from the model itself (e.g., PPL) or a task-specific scorer. This is an important feature. However, from the results in table 2, it seems like the proposed verifier is not good enough since the improvement over majority voting is quite marginal. I would suggest the authors to pay more attention to the general verifier and further improve it.
2. Some of the claims in the paper are over-claims. For example, Many existing works are indeed working on addressing the middle errors hence the first claim in the abstract might not be accurate. Also, from the view of a higher level, all the search algorithms (including self-consistency, beam search, ordinary tree search, MCTS) are all trying to alleviate the error caused by the greedy decoding. The key difference is the signal used to conduct the sampling. Hence, one should directly compare these methods instead of adding them together. For the extreme case, one should be able to combine all of these methods together but that is not helpful in a real system since it is too heavy.

---

> ### Author Rebuttal · Authors · 2024-05-30
>
> Thanks for the valuable suggestions!
>
> > Overclaim
>
> Thanks for bringing this up, we have noticed the overclaims and will revise them in the next version.
>
> > Missing comparisons
>
> We have compared recent methods involving search and verification [1,2] in Sec 5.2. For tree search and MCTS methods, we copy GSM8K results from [3,4]. [3] shows 40% accu on LLaMA-33b, comparable to LLaMA2-13b, while DBS reaches 43%. [4] shows 90% accu on GPT-4, while DBS reaches 92%.
>
> [1] Self-evaluation guided beam search for reasoning, NeurIPS 2024.
>
> [2] Deductive verification of chain-of-thought reasoning, NeurIPS 2024.
>
> [3] Reasoning with language model is planning with world model, arXiv 2023.
>
> [4] Tree of thoughts: Deliberate problem solving with large language models, NeurIPS 2024.
>
> > Performance on self-consistency
>
> We have shown **universal** improvements across different model scales and reasoning task genres. This is beyond reaching a SOTA on one single task, meaning that DBS robustly and generally works. Table 2 shows significant improvements on single reasoning chains, some surpassing self-consistency. This indicates the effectiveness of DBS in different sampling strategies (greedy and self-consistency).
>
> > Methods should directly compare rather than add
>
> Agree. That’s why we design both single and multiple reasoning chain settings and also compare DBS with two SOTA methods in Sec 5. Single chain proves the effectiveness and the multiple one proves the consistency. Thus, the comparison in each setting is fair and proves DBS's effectiveness and consistency.
>
> As for your concern that the system is too heavy to use, combining DBS with SC adds no time, and accurate reasoning, although slow, catalyzes stronger models through synthesized data [1,2].
>
> [1] Algo: Synthesizing algorithmic programs with generated oracle verifiers, NeurIPS 2024.
>
> [2] Solving olympiad geometry without human demonstrations, Nature, 2024.
>
> > Big Bench Hard
>
> BBH is not specifically designed for reasoning tasks. Our performance on various reasoning tasks shows generality. Moreover, our method can be applied as a plug-in-and-play, as shown in Sec. 6.3, and can be used for BBH tasks requiring reasoning.
>
> > Hyperparameters
>
> For all models and baseline, we use the default generation setting, that is temp=1, topP=1, topK=50. For self-consistency, we sample reasoning paths 10 times and adopt majority voting. All hyperparameters are set equally across baselines. We will clarify this in the next version.

---

> > ### Comment · Reviewer_R7sC · 2024-06-02
> >
> > Thanks for the reply. However, I still tend to keep my original rating because the two main arguments do not convince me.
> >
> > First, "combining DBS with SC adds no time." This comment is not accurate. Increasing the beam size will significantly improve the KV caching and computing power of GPUs. Especially, in DBS, the definition of each step is not very clear to the model. A streaming inference engine cannot know which token to split and merge. Also, the communication between the LM and the verifier model will also significantly increase the delay.
> >
> > Second, BBH is designed to evaluate how well models can solve complex tasks and reasoning capability is one of their focuses. BBH is also a popular dataset used in many recent LLM evaluation packages.

---

> > ### Author Response · Authors · 2024-06-03
> >
> > Thanks for your prompt reply. We are glad that we have addressed your other major concerns about **performance on self-consistency** and **missing comparisons**. We now would like to clarify the other two arguments.
> >
> > > Inference Time
> >
> > Thanks for bringing the inference time up. Actually, “combining DBS with SC adds no time” is the response to your concern about “Methods should directly compare rather than add”. DBS can cache all finished reasoning paths during inference. Thus, users can directly apply self-consistency on the DBS-generated reasoning paths, which makes DBS+SC add no time compared to DBS alone. \
> > As for your concern on DBS increasing inference time, we admit that applying DBS indeed increases inference time and may be challenging to be deployed as a delay-sensitive system in the near future. Yet, being able to be used directly in the real-world systems is *one of the many forms of metric* to evaluate the scientific contributions. We would like to highlight again that although accurate reasoning is more time-consuming, it catalyzes stronger models [1,2,3,4]. The cited works all use *“time-consuming”* sampling methods, which synthesize accurate reasoning paths, and finally breeds more powerful models. Moreover, as we have discussed in the response to Reviewer 2vf7, we compare the DBS inference time with self-consistency, where the additional time cost is primarily introduced by the small verifier, and verifier inference time is 0.1x of LLM inference time. Ultimately, in terms of future impacts, we believe that accuracy is a critical factor for stronger models. \
> > Thanks again for pointing out the limitation from the immediate use perspective, we will add more discussions in terms of inference speed in the future revision.
> >
> > > Big Bench Hard
> >
> > Thanks for the suggestion. We respectfully suggest that BBH may not be indispensable for work focusing on reasoning ability. As you pointed out, “reasoning capability is one of their focuses.” Given the multitude of abilities involved, including natural language understanding, it can be challenging to evaluate **reasoning ability**. Moreover, in recent works focusing on improving reasoning ability [5,6,7,8,9,10], BBH is not necessarily evaluated to demonstrate the effectiveness of reasoning. We are happy to discuss more if you could possible consider elaborating more.
> >
> > Thanks again for your valuable suggestions and for being willing to discuss, let us know if you have any other questions.
> >
> > [1] Kexun Zhang, et al. Algo: Synthesizing algorithmic programs with generated oracle verifiers, NeurIPS 2024.
> >
> > [2] Trieu H. Trinh, et al. Solving olympiad geometry without human demonstrations, Nature, 2024.
> >
> > [3] Jiaxin Huang, et al. Large Language Models Can Self-Improve. EMNLP 2023
> >
> > [4] Zelikman Eric, et al. Star: Bootstrapping reasoning with reasoning. NeurIPS 2022
> >
> > [5] Ning Miao, et al. SelfCheck: Using LLMs to Zero-Shot Check Their Own Step-by-Step Reasoning. ICLR 2024
> >
> > [6] Terufumi Morishita, et al. Learning Deductive Reasoning from Synthetic Corpus Based on Formal Logic, PMLR 2023
> >
> > [7] Yuxi Xie, et al. Self-evaluation guided beam search for reasoning, NeurIPS 2024.
> >
> > [8] Zhan Ling, et al. Deductive verification of chain-of-thought reasoning, NeurIPS 2024.
> >
> > [9] Shibo Hao, et al. Reasoning with language model is planning with world model, EMNLP 2023.
> >
> > [10] Xidong Feng, et al. Alphazero-like tree-search can guide large language model decoding and training, 2023

---

### Official Review · Reviewer_iM3N · 2024-05-12

**Rating:** 7
**Confidence:** 3
**Ethics Flag:** 1

**Summary:**

This paper proposes Deductive Beam Search (DBS), which integrates chain-of-thought (CoT) and deductive reasoning with step-wise beam search for LLMs. To train such verifier, the authors propose labor-free data construction methods by replacing numbers in the equations with randomly generated ones and randomly selecting reasoning steps. The experiments show the effectiveness of the proposed method and the paper is quite easy to understand.

**Reasons To Accept:**

* The proposed method combines deductive reasoning with beam search exploration
* Propose a labor-free data construction method for training the verifier, which generates grounding errors and logic errors.

**Reasons To Reject:**

* Lack of further analysis of different reasoning steps

---

> ### Author Rebuttal · Authors · 2024-05-30
>
> Thanks for the valuable suggestions!
> > Further analysis of different reasoning steps
>
> Thanks for the suggestion, we show results w.r.t different reasoning steps in the table below.
> It shows that DBS can maintain higher accuracy even when the length of the reasoning process is up to 15 steps.
> We will add them to the next version.
>
> |DBS CoT    |||Greedy CoT|||
> |-----------|----------|----------|----------|----------|----------|
> |Step Length|Accuracy  |Number |Step Length|Accuracy|Number    |
> |3          |92.8      |180    |3          |83.4    |241       |
> |4          |84.8      |322    |4          |74.2    |337       |
> |5          |80.3      |285    |5          |66.6    |234       |
> |6          |70.9      |189    |6          |60.5    |177       |
> |7          |68.6      |101    |7          |51.1    |92        |
> |8          |67.6      |86     |8          |62.5    |48        |
> |9          |68.2      |55     |9          |59.5    |42        |
> |10         |67.1      |27     |10         |52.9    |34        |
> |11         |52.6      |19     |11         |47.6    |21        |
> |12         |100       |8      |12         |48.3    |12        |
> |13         |57.6      |17     |13         |60      |10        |
> |14         |66.6      |6      |14         |50      |10        |
> |15+        |50        |2      |15+        |38.2    |9         |

---

### Official Review · Reviewer_2vf7 · 2024-05-16

**Rating:** 5
**Confidence:** 4
**Ethics Flag:** 1

**Summary:**

In this paper, the authors propose Deductive Beam Search (DBS) to alleviate accumulated errors in LLM CoT reasoning. Specifically, it integrates the step-wise beam search method into LLM CoT and designs a deductive verifier to verify intermediate reasoning step.
Experiments results show that DBS improves the base performance of LLMs on various task and can detect diverse and subtle reasoning errors and robustness on different model scales, proving its superiority over previous methods.

**Questions To Authors:**

The idea in DBS is straightforward, and the results show that it is a promising approach. I have some questions regarding the DBS method and experiments:

1. The paper only using two relatively weak LLMs to evaluates DBS on eight datasets. Considering the submission time, the state-of-the-art accuracy should have already surpassed the results reported in the paper, which may undermine the persuasiveness of the work.
2. DBS categorizes the errors in the reasoning process into several types, but these do not cover all the errors encountered in real-world scenarios (which is also mentioned in the paper). If the reasoning process is deductive reasoning, DBS aligns with expectations. However, many real-world reasoning tasks also involve inductive and abductive reasoning. Can you discussing DBS's potential limitations or extensions on these other forms of reasoning?
3. What is the model's strategy for handling scenario where multiple errors coexist simultaneously in one intermediate process ?

**Reasons To Accept:**

1. The integration of step-wise beam search and a deductive verifier with CoT reasoning is a novel and effective approach to mitigating accumulative errors in multi-step reasoning.
2. Comprehensive experiments on diverse reasoning tasks and LLM scales demonstrate the broad applicability and effectiveness of DBS.
3. The verifier analysis provides good insights into its error detection capabilities, robustness, and impact on the deducibility of generated reasoning paths.

**Reasons To Reject:**

1. Paper does not provide a detailed analysis of how different beam search hyperparameters impact this trade-off and the final performance.
2. The computational efficiency of DBS compared to baselines is not discussed in depth. Given the additional overhead of beam search and verification, an analysis of the inference time would be informative.
3. The paper focuses on the deductive reasoning paradigm, but many real-world reasoning tasks also involve inductive and abductive reasoning. Discussing DBS's potential limitations or extensions to these other forms of reasoning could broaden its impact.
4. There are significant differences among various types of reasoning tasks. DBS is more effective in Arithmetic Reasoning task. The paper lacks analysis and discussion in this regard.

---

> ### Author Rebuttal · Authors · 2024-05-30
>
> Thanks for the valuable suggestions!
>
> > Hyperparameters
>
> We have conducted such analysis in Appendix A in the current submission and will move it to the main part in our revised version.
> In terms of the final performance, our verifier can select gold reasoning steps with an 86% probability (Sec 6.1).
>
> > Inference time
>
> While DBS increases inference time, the additional time cost is primarily introduced by the small verifier, as other sampling methods like self-consistency also require more LLM inference time. Setting all things equal, verification costs less than 0.1x of generation, making our method 1.1x the cost of self-consistency.
> Despite its relatively slow speed, DBS outperforms baselines in performance and **token cost** in Sec 5.2.
> We will discuss about inference speed in the next version.
>
> > Limitations & Extensions
>
> Thanks for the suggestion, we discussed how to further improve the performance on tasks requiring other abilities as an extension, i.e. commonsense reasoning tasks, in Sec 6.3. Although DBS is designed for deductive reasoning, it benefits other tasks on deductive reasoning steps.
> Indeed, our data synthesis strategy narrows the scope to deductive reasoning, but DBS, as a plug-in-and-play, will not harm the performance of other tasks. Discussing further, one might apply our proposed data synthesis strategy to train a powerful general verifier for all reasoning types.
>
> > Model Choice
>
> In terms of open-sourced LLM, LLaMA2 is arguably one of the most powerful open-source model families that covers a full range of model scales to showcase DBS's generality. In terms of closed-sourced LLM, the expense of full evaluation on gpt-4 is around **10k** dollars, which is too much with limited budget.
> To further prove the effectiveness of DBS, we randomly sample 100 samples from the datasets and evaluate DBS on Mixtral and gpt-4. DBS shows promising improvements as well, especially considering Mixtral combined with DBS is able to compete with the gpt-4.
>
> |||GSM8K|StrategyQA|Coin|
> |------------|------|-----|----------|----|
> |Mixtral|Greedy|73|70|78|
> ||SC|84|74|75|
> ||DBS|85|74|84|
> ||DBS+SC|91|75|77|
> |gpt-4-turbo|Greedy|92|74|100|
> ||SC|94|78|100|
> ||DBS|94|76|100|
> ||DBS+SC|95|78|100|
>
> > Multiple errors
>
> If multiple errors happen in sampling, DBS takes the step that scores the highest.
> If multiple errors happen in one reasoning chain, DBS assumes all previous reasoning steps are true so that they can be used as premises for the following steps.

---

### Decision · Program_Chairs · 2024-07-10

**Decision:**

Accept

**Comment:**

The reviewers agreed this is an important research direction and interesting algorithm. Most of the concerns were about baselines and exact sets of experiments, etc. However, the authors were able to directly answer all of the concerns and clarify confusions to my satisfaction, even if one of the reviewers didn't agree.